# Nano- and Crystal Engineering Approaches in the Development of Therapeutic Agents for Neoplastic Diseases

Emmanuel M. Kiyonga [1,†], Linda N. Kekani [1,†], Tinotenda V. Chidziwa [1,†], Kudzai D. Kahwenga [2], Elmien Bronkhorst [2], Marnus Milne [1], Madan S. Poka [1], Shoeshoe Mokhele [1] and Bwalya A. Witika [1,*]

[1] Department of Pharmaceutical Sciences, School of Pharmacy, Sefako Makgatho Health Sciences University, Pretoria 0208, South Africa; kiyonga.emmanuel.emmanuel@gmail.com (E.M.K.); linda97kekana16@gmail.com (L.N.K.); tinotendac1@gmail.com (T.V.C.); marnus.milne@smu.ac.za (M.M.); madan.poka@smu.ac.za (M.S.P.); shoeshoe.mokhele@smu.ac.za (S.M.); patrick.demana@smu.ac.za (P.H.D.)

[2] Department of Clinical Pharmacy, School of Pharmacy, Sefako Makgatho Health Sciences University, Pretoria 0208, South Africa; daisykudzai@yahoo.com (K.D.K.); elmien.bronkhorst@smu.ac.za (E.B.)

* Correspondence: bwalya.witika@smu.ac.za

† These authors contributed equally to this work.

**Abstract:** Cancer is a leading cause of death worldwide. It is a global quandary that requires the administration of many different active pharmaceutical ingredients (APIs) with different characteristics. As is the case with many APIs, cancer treatments exhibit poor aqueous solubility which can lead to low drug absorption, increased doses, and subsequently poor bioavailability and the occurrence of more adverse events. Several strategies have been envisaged to overcome this drawback, specifically for the treatment of neoplastic diseases. These include crystal engineering, in which new crystal structures are formed to improve drug physicochemical properties, and/or nanoengineering in which the reduction in particle size of the pristine crystal results in much improved physicochemical properties. Co-crystals, which are supramolecular complexes that comprise of an API and a co-crystal former (CCF) held together by non-covalent interactions in crystal lattice, have been developed to improve the performance of some anti-cancer drugs. Similarly, nanosizing through the formation of nanocrystals and, in some cases, the use of both crystal and nanoengineering to obtain nano co-crystals (NCC) have been used to increase the solubility as well as overall performance of many anticancer drugs. The formulation process of both micron and sub-micron crystalline formulations for the treatment of cancers makes use of relatively simple techniques and minimal amounts of excipients aside from stabilizers and co-formers. The flexibility of these crystalline formulations with regards to routes of administration and ability to target neoplastic tissue makes them ideal strategies for effectiveness of cancer treatments. In this review, we describe the use of crystalline formulations for the treatment of various neoplastic diseases. In addition, this review attempts to highlight the gaps in the current translation of these potential treatments into authorized medicines for use in clinical practice.

**Keywords:** cancer; co-crystals; nanocrystals; nano co-crystals; crystal engineering; nanoengineering; regulatory aspects

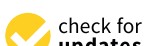



## 1. Introduction

Cancer is one of the leading causes of death in the world, with an estimated 10 million cancer-related deaths recorded for the year 2020 [1]. The risk of cancer for people that are aged between 0 and 74 years is 20.2%, with estimated prevalence of 22.4% in men and 18.2% in women [1]. In 2018, 18 million cases of cancer were diagnosed, with lung and breast cancer cases contributing 2.09 million each, and 1.28 million cases of prostate cancer [2]. A total estimate of 1,918,030 new cancer cases and 609,360 cancer deaths were projected for the United States for the year 2022 with 350 daily deaths from lung cancer alone [3]. Breast

cancer remains the leading cause of cancer-related deaths in females [4] and continues to increase despite the availability of treatment options. Several antineoplastic agents are administered via the intravenous (IV) route and this comes with challenges of side effects and cytotoxicity [5,6] as well as inconveniences associated with IV infusion [7], such as regular visits to the hospital, pain, and extravasation [8,9]. These can reduce patient adherence to treatment and contribute to treatment failure. The route of administration depends mainly on the physicochemical properties of the drug, hence IV route is the preferred route for drugs with low bioavailability [10]. The majority of anti-neoplastic agents exhibit poor aqueous solubility and low bioavailability after oral administration [11].

Poor aqueous solubility remains a major challenge for many anticancer API as their bioavailability is reduced by poor dissolution rates leading to a decrease in pharmacological activity [12,13]. Many of these drugs exist as polycrystalline materials which could result in different physicochemical-associated physical stability and solubility, thereby compromising their therapeutic outcome [14]. Crystal engineering is a discipline that involves manufacturing drug molecules with new crystal structures and properties using intermolecular interactions to improve physicochemical properties [15,16]. This technology can be used in designing new molecules by re-structuring crystalline materials for purposes of improving solubility bioavailability, stability, hygroscopicity, compressibility, and photoluminescence [15,17,18]. This can be achieved through polymorphic selection, the formation of amorphous, co-amorphous, and co-crystalline materials, which all possess the potential of enhancing products' pharmaceutical performance [17].

Co-crystallization is a method that has been utilized to improve the aqueous solubility of active pharmaceutical ingredients (API) that exhibit poor aqueous solubility [18]. The ability of co-crystals to enhance apparent solubility, hence dissolution of APIs with low solubility and bioavailability of class II and IV drugs, has attracted the attention of many researchers focusing on drug delivery systems [19,20]. Co-crystals are supramolecular complexes of an API and a co-crystal former (CCF), usually in stoichiometric ratios, held together by non-covalent interactions such as van der Waals forces, π–π stacking, and hydrogen bonds in a crystal lattice [21]. Co-crystals have been used to improve treatment of different diseases such as HIV, heart failure, motion sickness, and different bacterial infections [22–26]. In many instances, the choice of CCF may be another API, resulting in multidrug co-crystals with potential additive or synergistic effects [27].

Nanocrystals can deliver APIs with various physicochemical properties and varying degrees of hydrophilicity [28]. Nanosystems increase the concentration of the drug selectively to the target site, while reducing side effects associated with wide drug distribution. Therefore, nanosystems were found to reduce toxic effects brought by antineoplastic agents [29]. These systems are able to identify the target tissue [30], due to surface coating, labeling, or modifications, enhancing selective uptake of the drug by target tissue [31]. With decreased size and increased surface area-to-volume ratio while still maintaining the structure of the nanocrystals [32], nano co-crystals enable further increase in drug dissolution by combining the advantages of co-crystals with those of nanocrystals.

The ability of crystalline formulations to improve drugs' physicochemical properties such as drug solubility and stability [33], combined with the ability of nanocrystalline formulations to target neoplastic tissue [30,31], make co-crystallization and nanonization ideal strategies to improve the antineoplastic activity of cancer drugs and make cancer treatments effective.

In this review, focus is placed on highlighting the use of crystalline formulations in the treatment of various cancer types, while identifying the gaps in the current translation of these potential treatments into authorized medicines for use in clinical practice.

## 2. Pathology of Neoplastic Disease

The proliferation of normal healthy cells is strictly regulated, with stimulatory and inhibitory signals in a delicate balance. For cancer cells to develop, a physical, chemical, or biological agent must damage the cell and cause an alteration that is subsequently

propagated during cell division. The exact mechanism by which cancers occur are, however, incompletely understood. These alterations may lead to unlimited growth, invasion, and metastases [34].

Carcinogenesis is postulated as a multistage process that is genetically regulated. The first step includes exposure of normal cells to carcinogens, which may include chemical compounds like asbestos and benzene, and drugs and hormones used for therapeutic purposes. Furthermore, physical agents such as radiation and ultraviolet light and biologic agents like viruses or pollutants can also act as carcinogens. Hereditary factors like age and gender may also play a role. Carcinogens produce genetic alterations that can result in irreversible cellular changes. The changed cell has an altered response to its environment and provides selective growth, resulting in a clonal population of cancer cells. This is followed by promotion, where carcinogens alter the environment to favor growth of the altered cell population. The last step follows, namely conversion or transformation, where the altered cell becomes cancerous. Depending on the specific cell, this process may happen over 5–20 years before development of a clinically detectable cancer [35,36]. Progression is the final stage in cancer development and involves further genetic alterations that lead to increased cell proliferation. This stage includes invasion into local tissues as well as the development of metastases [35]. Female breast cancer and lung cancer account for the highest number of new cases, while lung cancer contributes to the highest number of deaths globally, as depicted in Figure 1 [37].

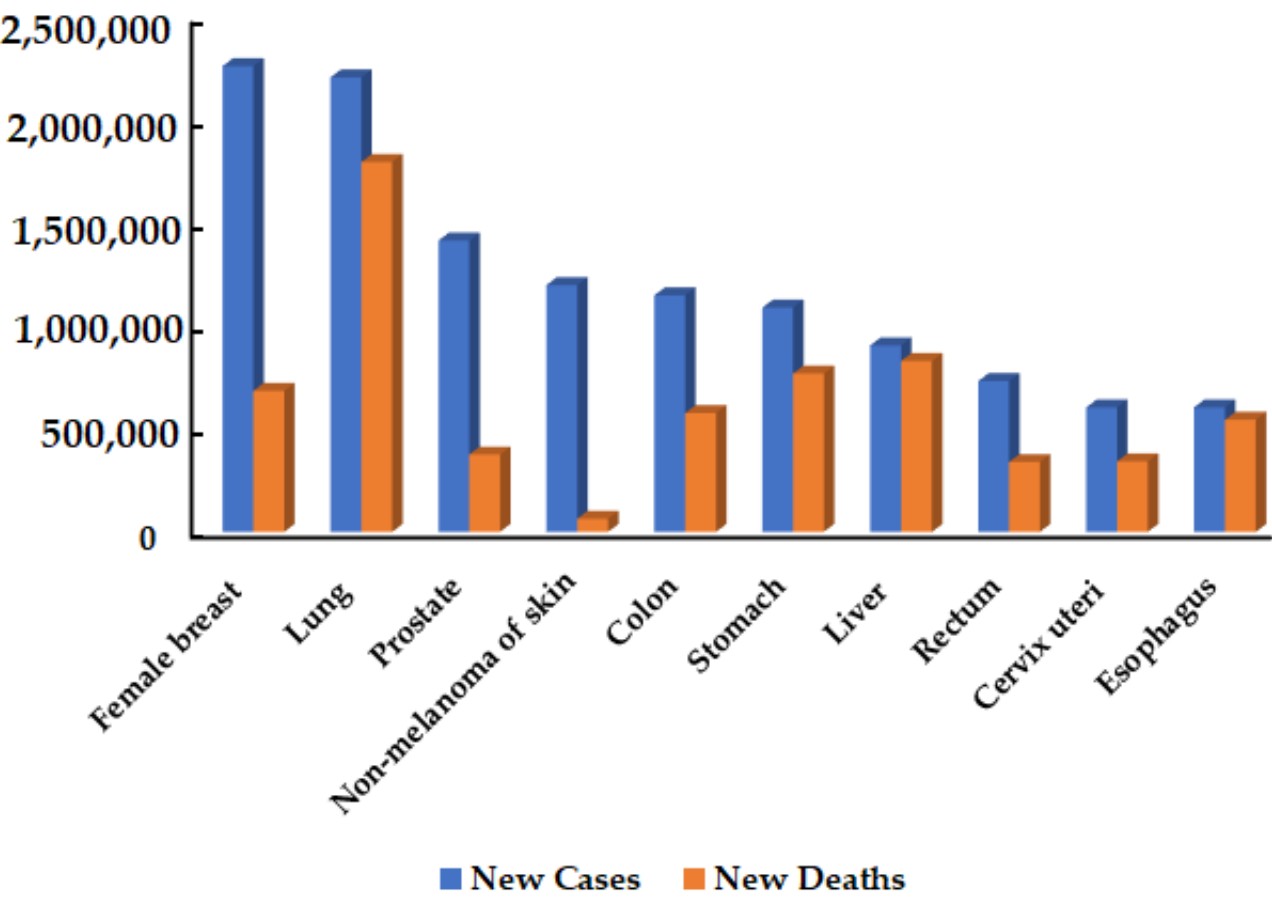

**Figure 1.** Predominance of different cancer types in 2020 (replotted from data obtained from [37]. Copyright 2022, copyright holder John Wiley and Sons, New Jersey).

*General Treatment Modalities*

Traditionally, three modalities are employed to treat cancer, namely surgery, radiation therapy, and systemic anticancer agents or chemotherapy, targeted drugs, and biologic therapies. These modalities are typically given sequentially or concurrently to treat specific cancers.

The goals of treatment depend on the cancer stage and patient factors, such as comorbidities. When an anticancer agent is administered to patients with local or regional disease, the treatment can be labeled as curative therapy when it can completely cure the cancer, or as palliative therapy when cancer has metastasized to distant sites. Palliative therapy will be employed to slow the progression of cancer and prolong survival by months to years

The study of cancer-growth forms the foundation for many of the basic principles of modern chemotherapy [35]. A chemotherapeutic agent is given as part of a combination regimen where agents with different mechanisms and toxicities are given together.

Currently, more than 100 anti-cancer chemotherapeutic agents are marketed. These medications are divided into six classes, illustrated in Table 1.

**Table 1.** Types of chemotherapy medication [38].

| Medication Class | Mechanism of Action | Examples | Clinical Application |
|---|---|---|---|
| Alkylating agents | Act directly on DNA causing cross-linking of DNA strands, abnormal base pairing, or DNA strand breaks, thus preventing the cell from dividing. | Chlorambucil Cyclophosphamide Cisplatin Carboplatin | Treatment of slow-growing cancers |
| Nitrosoureas | Slow down or stop enzymes that help repair DNA. | Carmustine Lomustine | Malignant gliomas, brain metastases of different origin, melanomas Hodgkin disease |
| Anti-metabolites | Replace natural substances as building blocks in DNA molecules, thereby altering the function of enzymes required for cell metabolism and protein synthesis. | Fluorouracil Methotrexate Fludarabine | Leukemias Cancers of the breast, ovary, and the intestinal tract |
| Plant alkaloids and natural products | Act specifically by blocking the ability of a cancer cell to divide and become two cells by inhibiting the dynamics of microtubules by binding to β-tubulins. | Vincristine Paclitaxel Topotecan | Various forms of cancer |
| Anti-tumor antibiotics | Act by binding with DNA and preventing ribonucleic acid synthesis, a key step in the creation of proteins, which are necessary for cell survival. Cause the strands of genetic material that make up DNA to uncoil, thereby preventing the cell from reproducing. | Bleomycin Doxorubicin Mitoxantrone | • Acute lymphocytic leukemia (ALL) • Acute myeloid leukemia (AML) • Breast cancer • Lymphoma (both Hodgkin's and non-Hodgkin's) • Variety of metastatic cancers (breast, bladder, bone sarcomas, lung, ovarian, neuroblastoma, stomach cancer) |
| Hormonal agents: • Corticosteroid hormones • Sex hormones | Induce apoptosis, or programmed cell death, in certain lymphoid cell populations Competes with or block hormone receptors, inhibiting hormone-dependent cell-growth. | Prednisone Dexamethasone Tamoxifen Leuprolide | Leukemia, multiple myeloma, and lymphoma Control the growth of breast, uterine and prostate cancers |
| Biological response modifiers | Strengthen the bodies' immune system to fight the growth of cancer. | Herceptin and Avastin Erbitux and Rituxan | Leukemia, lymphoma, melanoma, breast cancer, bladder cancer |

Most chemotherapy agents target rapidly proliferating cells at one or more phases of the cell cycle [34,38]. Chemotherapeutic agents typically interfere with the cellular synthesis of DNA, ribonucleic acid, and proteins. Agents can prevent the unwinding of DNA and thus inhibit protein synthesis (alkylators) or inhibit enzymes involved in the synthesis of DNA and proteins (antimetabolites) [39].

## 3. Co-Crystals

Co-crystals are defined as crystalline single-phase solid materials composed of two or more different molecular compounds and/or ionic compounds, that are electrically neutral and are generally in a stoichiometric ratio [40,41]. These molecules are solids at room temperature, and they rely on the hydrogen-bonded assemblies between neutral molecules of the API and the CCF [41–44].

Co-crystals can improve the physicochemical properties of the drug, such as processibility, melting point, friability, permeability, and solubility of the poorly water-soluble drug, and hence its bioavailability. They are less prone to phase changes and can be preserved in a humid environment due to their crystallinity, and are often resistant to drug processing such as wet granulation and tableting [42,43]. Co-crystals have a long-range order; thus, they are more thermally stable than amorphous solids which have a short-range order. The amorphous solid-state form has higher Gibbs free energy, internal energy, specific volume, solubility, and thus dissolution rate compared to co-crystals. This energy results in decreased physical and chemical stability thus creating a possibility of the amorphous form recrystallizing during storage [44,45]. Furthermore, co-crystals can reduce the dose and adverse effects of the API without altering the chemical composition of the drug, thus improving patient compliance which is usually compromised because most patients do not complete their treatments [34,44,46,47].

Co-crystallization, the process of forming co-crystals, makes use of non-covalent bonds such as π–π stacking, van der Waals forces, and hydrogen bonding, without the transfer of hydrogen ions to form salts as depicted in Figure 2 [48,49].

Molecules that are already polymorphic and molecules that can adopt other packing patterns, while still fulfilling the needs of the hydrogen-bond acceptor or donor existing on the two components, should be employed during co-crystallization. Therefore, it can be inferred that the structure of the CCF has a significant impact on the co-crystal [46,47].

Co-crystals are formed using different methods and the most used approaches are classified as the solid-state method and the solution-based method. Crystal properties such as crystal size and crystal morphology, among others, determine the type of co-crystallization method to be used [41,42,50].

Solid-state methods involve techniques such as neat grinding and hot-melt extrusion. These methods use little or no solvent for co-crystallization and the co-crystal formation is usually forced through shear mixing by melting and re-solidification [47,51]. This method is usually used because it is regarded as a "green technique" as it is a single-step process that has few by-products and avoids the use of solvent, and thus the possible formation of solvates and hydrates. However, this lacks control over the crystal process, and it cannot be used for thermos-labile API [47,52].

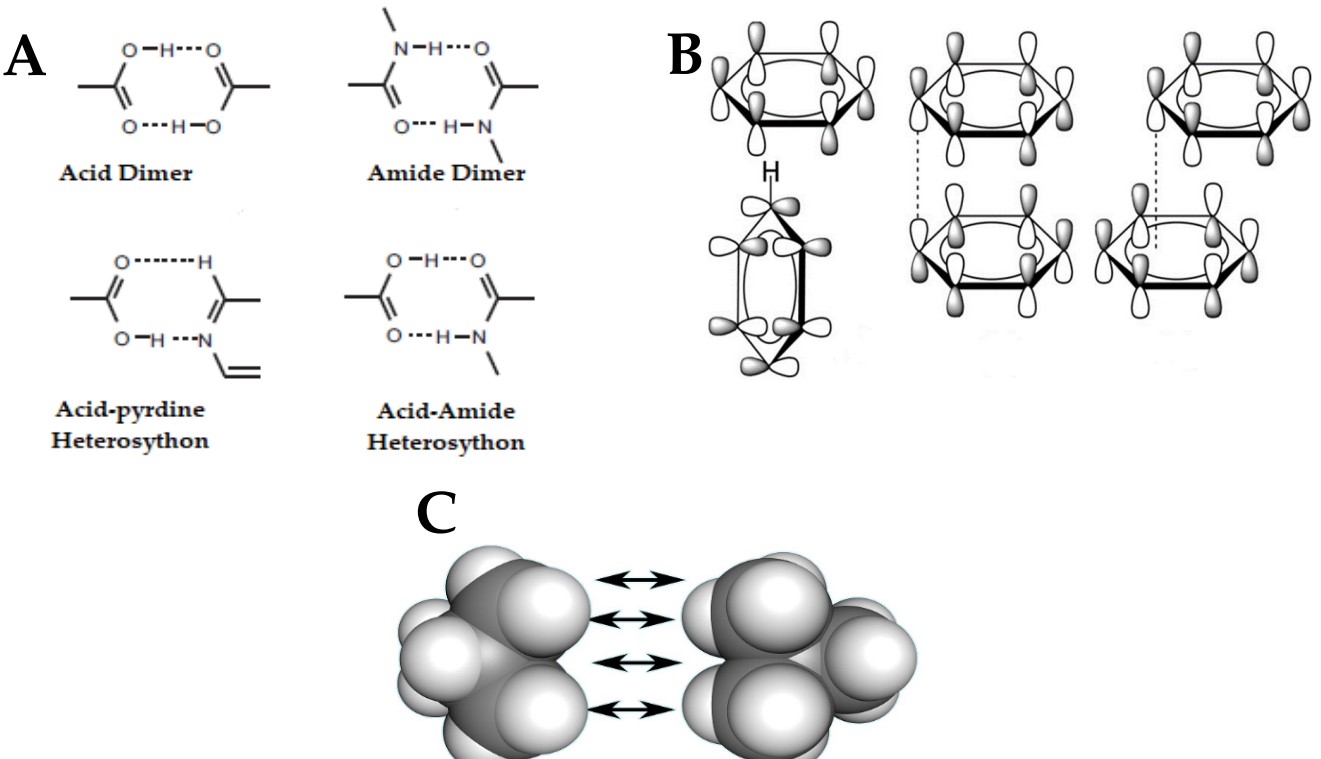

**Figure 2.** Depiction of the forces involved in co-crystal formation: (**A**) Hydrogen bonding (reproduced from [51]. Copyright 2022, copyright holder Elsevier, Amsterdam); (**B**) π–π stacking (reproduced from [52]. Copyright 2022, copyright holder Elsevier, Amsterdam); (**C**) van der Waals interactions (reproduced in accordance with https://creativecommons.org/licenses/by/4.0/, accessed on 23 June 2022).

Although equal solubility is required for solution-based co-crystallization to avoid the least soluble component from precipitating out and causing crystal separation, this does not ensure success. However, the use of polymorphic compounds is said to be beneficial. Solution-based co-crystallization utilizes solvents and supersaturation as the required driving force. The solvent has the potential to modify the intermolecular interactions between the API and CCF. Techniques such as evaporation, slurry, and cooling crystallization are examples of this method [47–53]. Recently, more research is being done on co-crystallizing APIs of some anticancer, antiretroviral, and antibacterial drugs [53–55].

Bhatt et al. demonstrated that one such multidrug co-crystal could be manufactured by co-crystallizing the two nucleoside reverse transcriptase inhibitors, zidovudine and lamivudine, to produce a hydrate co-crystal for HIV therapy [24]. Similarly, Entresto® (Otsuka Pharmaceutical Co., Ltd., Japan), a medication used in the treatment of heart failure, is a combination of valsartan, an angiotensin II inhibitor, and sacubitril, a neprilysin inhibitor. The resultant co-crystal has led to an increased bioavailability of valsartan [56]. In the treatment of bacterial infections, the co-crystal formed by co-crystallizing amoxicillin and clavulanate exhibited increased antibacterial activity when compared to amoxicillin alone, due to β-lactamase inhibition attributed to clavulanate [22] Similarly, co-crystallization of isoniazid and pyrazinamide resulted in synergistic effect for tuberculosis treatment [57].

*Anticancer API Co-Crystals*

Co-crystals have found a wide array of use in the pharma industry including in anticancer therapy. In a study by Jubeen et al., it was observed that the co-crystal forms of 5 fluorouracil (5-FU), an anticancer drug, had different physicochemical properties depending on the CCF used [58]. The CCFs affect the API differently depending on their

individual properties such as anticancer, anti-inflammatory, and antioxidant activities. Solid-state grinding and slow evaporation methods were performed with acetone as a solvent for all the co-crystallization experiments. All formed co-crystals showed increased growth inhibition potential and anticancer activity compared to the main API, and 5-FU-cinnamic acid was found to be the most potent anticancer agent compared to the other co-crystals in vitro because of the synergistic effect of both drugs, as cinnamic acid itself has anticancer potential [58,59].

An investigation was done on the use of furosemide and mefenamic acid as CCF of erlotinib and gefitinib, respectively [60]. Two solvents were used, n-butanol was used for the erlotinib–furosemide co-crystal, and methanol–acetonitrile for the gefitinib–mefenamic acid co-crystal. Both co-crystals were created using dry and liquid assisted grinding, followed by solvent evaporation. Solubility and dissolution studies revealed that the stable form of gefitinib was more soluble than the gefitinib–mefenamic acid co-crystal, and the solubility of mefenamic acid in the co-crystal was increased. The molecules of mefenamic acid in the co-crystal are weakly associated through O-H—-O hydrogen bonding with gefitinib acid and the π–π interactions hold the mefenamic acid very tightly in the polymorph as compared with the co-crystal. When compared to erlotinib alone, erlotinib–furosemide co-crystal had a lower solubility and dissolution rate. Furosemide's solubility, like that of mefenamic acid, increased because the furosemide molecules are densely packed and thus are stronger in the co-crystal. These drug–drug co-crystals demonstrated improved thermal stability, density, and crystal packing [60].

Similarly, 5-FU was co-crystallized with a CCF with anti-cancer potential, nicotinamide. It was shown that the 5-FU-nicotinamide co-crystal had enhanced inhibitory activity and better anticancer effect in comparison to 5-FU. The authors attributed this decrease to be possibly from nicotinamide possessing antioxidant activity

Lastly, a conclusion was made that 5-FU-nicotinamide had enhanced solubility compared to the 5-FU alone, the logP value of 5-FU was higher than that of the co-crystal [61]. This showed that the co-crystal was more hydrophilic and had lower membrane permeability than the API. The in vitro and in vivo studies showed that the co-crystal did have increased bioavailability and anticancer activity. Further showing a decrease in acute toxicity. Although the co-crystal had enhanced solubility, bioavailability, and absorption compared to the API, the co-crystal and the API still had the same efficacy in vivo.

In another study conducted by Nicolov et al. [62], betulinic acid was co-crystallized with ascorbic acid. The co-crystallization was performed using the hydrothermal technique utilizing isopropyl alcohol as the solvent. The co-crystal formed exhibited increased cytotoxic effects due to the additive pharmacological effect, particularly towards murine melanoma cells lines, while preserving the API selectivity.

The in vitro study conducted by Aakeroy and Forbes proved that the solubility of the API, hexamethylenebisacetamide, could be improved without changing the chemical structure of the API. Dicarboxylic acids, also known as diacids, and ethanol as the solvent were used in the synthesis of the co-crystals. The diacids used were succinic acid, adipic acid, suberic acid, sebacic acid, and dodecanedioic acid. It was observed that the co-crystals manufactured with shorter-chain diacids had enhanced aqueous solubility relative to the API and the longer-chain diacids. The change in solubility was attributed to shorter-chain diacids being more polar and less hydrophobic in nature compared with the longer-chain diacids [41].

The summary of co-crystal studies for neoplastic diseases as depicted in Table 2.

**Table 2.** A summary of findings derived from co-crystal studies in the treatment of neoplastic diseases.

| API | CCF | API-CCF Interaction | Method of Preparation | In Vitro Model | In Vivo Model | Result | Ref. |
|---|---|---|---|---|---|---|---|
| 5-FU | Benzoic acid | Hydrogen bonding | Neat grinding and slow evaporation | MTT assay using human colorectal cancer cell (HCT 116) | - | Increased anticancer activity | [58] |
| | Cinnamic acid | Hydrogen bonding | Neat grinding and slow evaporation | MTT assay using human colorectal cancer cell (HCT 116) | - | Increased anticancer activity | [58] |
| | Malic acid | Hydrogen bonding | Slow evaporation and neat grinding | MTT assay using human colorectal cancer cell (HCT 116) | - | Increased anticancer activity | [58] |
| | Nicotinamide | Hydrogen bonding | Cooling technology | MTT assay and HE staining using human liver cell (BEL-7402/5-FU) | - | Enhanced antitumor activity Enhanced anticancer effect than 5-FU Solubility increased | [63] |
| | Nicotinamide | Hydrogen bonding and lone pair electron$-\pi$ stacking | Solvent evaporation and liquid phase-assisted grinding | MTT assay using HCT 116 tumor cells | Mice | The co-crystal had more anti-tumor properties than the 5-FU and solubility increased | [61] |
| | Succinic acid | Hydrogen bonding | Neat grinding Slow evaporation | MTT assay using human colorectal cancer cell (HCT116) | | Increased anticancer activity | [58] |
| Betulinic acid | Ascorbic acid | Hydrogen bonding | Hydrothermal method | Alamar blue assay and MTT assay using murine melanoma cells, human breast cancer (MCF-7,MDA-MB-231) cells, HaCat cells and cervical cancer (HeLa) | | Higher cytotoxic activity Enhanced solubility and bioavailability | [62] |
| Hexamethyl-enebisacetamide | Dicarboxylic acids | Hydrogen bonding | Solvothermal synthesis | Lung cancer cells | | Diacids with longer chains led to extremely low solubility | [64]. |

**Table 2.** *Cont.*

| API | CCF | API-CCF Interaction | Method of Preparation | In Vitro Model | In Vivo Model | Result | Ref. |
|---|---|---|---|---|---|---|---|
| Nandrolone | 3-amino-1,2,4-triazole | Hydrogen bonding | Solution reflux | MTT assay using cervical HeLa cells and 3T3 fibroblast cell line | | Non-cytotoxic against 3T3 normal fibroblast cell line | [65] |
| | Salicylic acid | Hydrogen bonding | Ball milling | MTT assay using cervical HeLa cells and 3T3 fibroblast cell line | | Co-crystal is a potent anticancer agent and is non-cytotoxic against 3T3 normal fibroblast cell line | [65] |
| Palbociclib | Orcinol | Hydrogen bonding | Solvent evaporation | MTT assay using human umbilical vein endothelial cell line (HUVEC) | Rats | Lower cytotoxicity compared to palbociclib Enhanced bioavailability and solubility increased | [66] |
| | Resorcinol | Hydrogen bonding | Solvent evaporation | MTT assay using human umbilical vein endothelial cell line (HUVEC) | Rats | Enhanced bioavailability and biosafety Enhanced absorption in rats and better plasma distribution | [66] |
| Tegafur | Isonicotinamide | Hydrogen bonding | Solvent evaporation and neat grinding | | | Solubility increased | [67] |

## 4. Nanocrystals and Nano Co-Crystals (NCC)

Formulations of novel nanocrystals and nano co-crystals have been developed with the aim of addressing poor solubility and poor bioavailability challenge that is found in approximately 60% of commercially available APIs [68].

Nanocrystals are defined as carrier-free submicron colloidal drug delivery systems with a mean particle size in the nanometer range, typically between 10–800 nm [69]. The main principle of nanocrystals and nano co-crystals (NCC) is reduction of particle size to nanoscale dimensions, thus increasing the surface area-to-volume ratio. Drug nanocrystal size reduction to the nanometer range modifies thermodynamic and kinetic properties, thus overcoming biopharmaceutical delivery challenges [69]. This results in the improved solubility, enhanced stability, increased adhesiveness to cell membranes, increased saturation velocity, and dissolution velocities of the drugs [32,70]. An increase in these factors results in an increase in oral bioavailability, penetration of drug molecules into the skin, and elimination of serious adverse reactions that result from cosolvents incapacitation [70].

Stabilizers are incorporated in the manufacture of nanocrystals and NCC to prevent agglomeration by providing steric and/or electrostatic repulsions [71]. Ionic surfactants, non-ionic surfactants, and polymers can be employed to stabilize nanocrystals. Ionic surfactants are typically implemented to maintain particles separated via electrostatic repulsion, whereas non-ionic surfactants and polymeric stabilizers can be used to create a steric barrier to prevent aggregation [72].

NCC are co-crystals in the nanometer range that consist of two or more molecules in a stoichiometric ratio [32]. NCC consists of two different techniques that include co-crystal preparation followed by the nanonization of the obtained co-crystal. The nanometer range gives them the ability of a greater surface area-to-volume ratio and the co-crystal nature further enhances the drug properties [73]. Much like co-crystals, NCC consist of an API and co-crystal former (CCF) assembled via non-covalent interactions or hydrogen bonds [73]. The CCF can consist of an acid-base salt, food additives, preservative, excipient, another API, minerals, antioxidants, amino acids, or vitamins. NCC are manufactured by utilizing the top-down and bottom-up techniques as in nanocrystal formulations [32].

NCC have superior properties compared to, and in some cases synergistic properties of, nanocrystals and co-crystals [69]. NCC advanced technologies have recently emerged as a strategy to further improve dissolution rates, bioavailability properties, mechanical properties, physical stability intrinsic solubility, melting points, bulk densities chemical stability, and hygroscopicity [73]. They also have reduced the systematic cytotoxicity of drugs, thus reducing their adverse effect in comparison to nanocrystals and co-crystals [74,75].

Nanocrystals and NCC can be manufactured by using top-down and bottom-up techniques with a suitable stabilizer [34]. Typically, top-down techniques make use of attrition forces to reduce micromolecules to nanosized crystals, while in bottom-up techniques molecular aggregation is prevented at specific size by the use of nucleation enhancers and aggregation disruptions as depicted in Figure 3 [32].

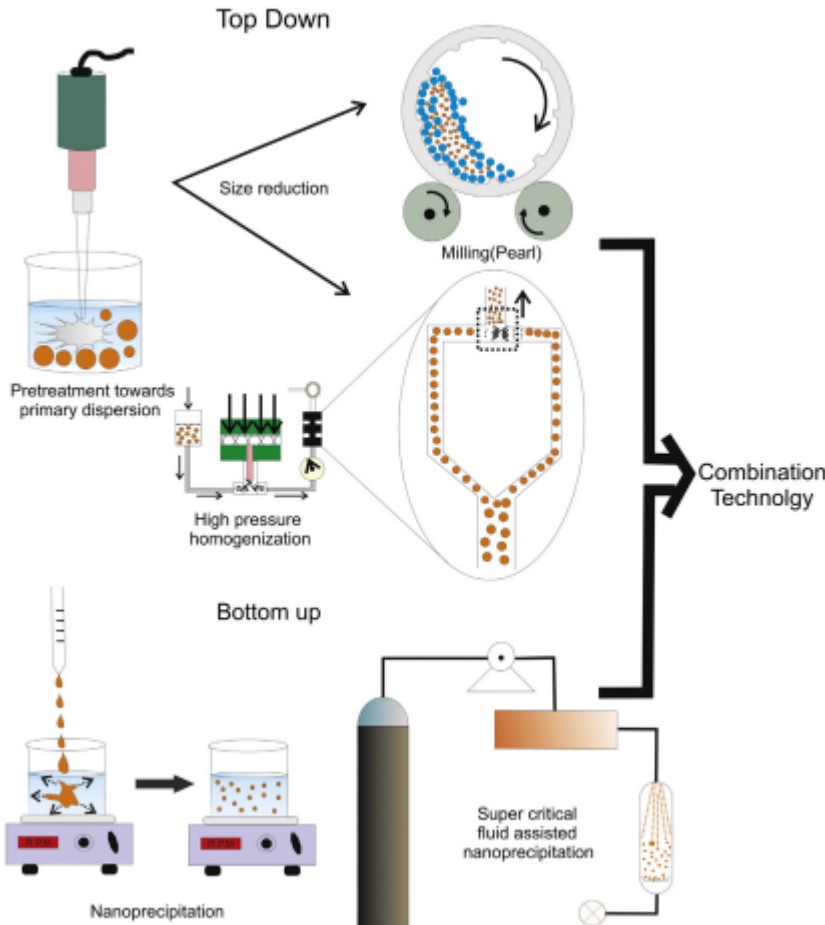

**Figure 3.** Schematic representation of the top-down method and bottom-up method (adapted from [76]. Copyright 2022, copyright holder Elsevier, Amsterdam).

Studies on anticancer nanocrystals have shown that nanocrystals can enhance drug targetability, drug loading, and drug distribution [77]. An improvement in these aspects results in a decrease in system cytotoxicity as more tumor cells are specifically targeted by the drug [77]. The basic principle of nanocrystal targeting depends upon the difference in the environments of the normal tissue and the cancer tissue. The targeting mechanisms examples include passive targeting, receptor-based targeting, enzyme response targeting, and pH-based targeting [78].

*Nanocrystals and Nano Co-Crystals in Cancer Treatment*

Nanocrystal and NCC technologies have been successfully used in the treatment of neoplasms. Bexarotene, classified as a synthetic retinoid activator, is indicated for the treatment of T-cell lymphoma (leukemia) and belongs to the BCS class II [79]. Its potential in numerous applications has been greatly hindered by its low bioavailability and low solubility [79]. Bexarotene nanocrystals were formulated using the precipitation method combined with microfluidization method with lecithin as a stabilizer [79]. The decrease of its particles to the nano range increased the surface area ratio leading to an increase in the dissolution rate [79]. Moreover, the increase in the dissolution rate resulted in an increase in bioavailability and a decrease in its side effects [79].

Camptothecin is a topoisomerase inhibitor indicated for the treatment of colorectal and lung cancers [80]. Camptothecin's (CPT) clinical indications are largely affected by its poor aqueous solubility, poor bioavailability, and presence of adverse side effects [80]. To curb these challenges, CPT nanocrystals were developed using hyaluronic acid due to their CD44

binding ability [80]. The hyaluronic (HA) coated acid was formulated by the antisolvent precipitation method [80]. The HA-coated CPT showed enhanced aqueous dispersion, drug loading efficiency, and stability improvements due to the hydrophilic HA and the nanosized particles [80]. The study revealed, in comparison to crude CPT, a decrease in toxic adverse effects on healthy cells, improvements in anticancer activity in vitro, and apoptosis-induced potency against overexpressed CD44 cancer cells was observed [80].

Etoposide (ETO) is an enzyme topoisomerase inhibitor that is largely utilized as a chemotherapeutic inhibitor for the treatment of ovarian cancer, lung cancer, breast cancer, testicular cancer, neuroblastoma, small lung cancer, and colon cancer [81]. It has very low water solubility and is often formulated together with a large proportion of cosolvents that lead to adverse effects [81]. Antisolvent precipitation method was utilized in preparing etoposide nanocrystal suspension [81]. The nanocrystals of ETO suspension demonstrated a sustained release drug profile kinetics in comparison to its marketed form [81]. The ETO nanocrystals in suspension also showed greater anticancer efficacy in vivo compared to the marketed form [81]. Furthermore, it also showed prolonged stability for months in its crystalline form. ETO nanocrystals proved to be more effective and safer than the marketed form [81].

In the study of baicalein–nicotinamide (BE-NCT) NCC, [72] formulated the nano co-crystal by using the high-pressure homogenization technique with poloxamer 188 as a stabilizer [68]. When the dissolution rate test was performed in vitro BE-NCT nano co-crystals had the highest dissolution rate in fasted stated simulated gastric fluid (FASSGF) compared to BE-NCT co-crystals, BE-nano crystals, and BE coarse powder [68]. In the dissolution, study BE-NCT NCC, BE-NCT co-crystals, and BE nanocrystals showed a 2.17 fold, 2.01 fold, and 1.74 fold increase in 360 min dissolution rates in comparison with BE coarse powder [68]. In vivo evaluations to investigate the pharmacokinetic factors were done after the oral administration of BE-NCT NCC, BE-nano crystals, BE-NCT co-crystals, and BE coarse powder. The results showed that an increase in AUC of 6.02-fold, 3.32-fold, and 2.87-fold for BE-NCT NCC, BE nanocrystals, and BE-NCT co-crystals (2), respectively [68].

Nicotinamide as a CCF was also incorporated in the fabrication of myricetin–nicotinamide NCC due to its good aqueous solubility properties [82]. The top-down and bottom-up methods were utilized in combination to fabricate these NCC [82]. The myricetin–nicotinamide NCC showed a higher dissolution rate than that of myricetin–nicotinamide co-crystal [82]. This was attributed to the large surface area-to-volume ratio produced after grinding the myricetin–nicotinamide co-crystal into a NCC [82]. These studies show that the formulation of recent novel NCC with CCF vitamins such as nicotinamide has potential uses in increasing bioavailability and dissolution rates of poorly soluble antineoplastic agents [82].

A novel codelivery NCC development was demonstrated by the formulation of paclitaxel–disulfiram NCC to prevent the development of multi-drug resistance (MDR) and enhance cytotoxicity in Taxol®-resistant cells during lung cancer treatment [83]. The paclitaxel–disulfiram NCC were prepared using the antisolvent precipitation method at a temperature of 4 °C with beta-lactoglobulin as a stabilizer [83]. The study revealed that paclitaxel–disulfiram NCC had a drug loading capacity of approximately 43% compared to 36% in paclitaxel nanocrystals [83]. In A549/Tax cells, the paclitaxel–disulfiram NCC uptake was 14-fold higher than that of paclitaxel nanocrystals. The disulfiram component resulted in an increase in uptake for the NCC. A decrease in MDR-1 expression in the NCC formulation was 2-fold compared to that of disulfiram [83]. This resulted in inactivation of the P-gp pump that influences multidrug resistance resulting in improved treatment efficacy [83]. Moreover, paclitaxel–disulfiram NCC resulted in a 5-fold rise in apoptosis and a 7-fold decrease in the $IC_{50}$ when compared with paclitaxel nanocrystals [83]. The decrease in $IC_{50}$ showed the potential reduction of side effects when paclitaxel–disulfiram NCC are utilized [83]. The characteristics of paclitaxel–disulfiram NCC can be used as a basis to further investigate the codelivery of NCC in overcoming MDR in chemotherapies [83].

Paclitaxel (PTX) and docetaxel (DTX) are taxane-based antineoplastic agents [84]. DTX is used to treat several malignancies, including lung cancer, prostate cancer, ovarian cancer,

and breast cancer [84]. The DTX's low water solubility (6–7 g/mL) is a significant drawback in its therapeutic application [84]. Docetaxel nanocrystals (DTX-NCs) were modified on their surface by incorporating human apo-transferrin (Tf) to enhance its cellular uptake and cytotoxicity of DTX [84]. The adsorption method was utilized to create the surface changes with Tf on DTX [84]. The DTX-NCs were formulated by using a bottom-up nanoprecipitation method using Tween® 80 as a stabilizer [84]. The A549 (human lung cancer) cell line was used in this work to perform an in vitro cytotoxicity investigation [84]. The DTX crystals modified by Tf had higher cytotoxicity (75%) in comparison to DTX-NCs (61%) and DTX pure (18.6%) after 48 h of incubation [84]. In quantitative cellular uptake analysis, Tf-DTX demonstrated a larger cellular uptake than DTX-NCs [84]. The study showed that Tf-DTX-NCs substantially enhanced the cellular uptake and cytotoxicity of DTX in vitro [84].

Oridonin (ORI) is classified as a ent-kaurene diterpenoid compound with anticancer properties and it has poor solubility properties [85]. ORI is a direct nucleolin inhibitor in various cancer cells that enhances cancer cell radio sensitivity [85]. In several studies, ORI has displayed antiproliferative and apoptosis-inducing effects on cells [85]. ORI nanocrystals were prepared by the antisolvent precipitation method with polyvinyl pyrrolidone K30 as a stabilizer [85]. The aim of formulation ORI nanocrystals was to improve their bioavailability by enhancing dissolution rate and solubility [85]. The intestinal barriers were modeled using Madin-Darby canine kidney (MDCK) cells because of their mucus film, similar polarization, and tight junctions to epithelial monolayers [85].

Moreover, ORI-NCs with fluorescent probe DiO loaded were synthesized to examine the transmembrane pathway and assess transcytosis on MDCK cells [85]. The results of the investigation indicated that the dissolution rate of ORI-NCs was much higher than that of the pure ORI in approximately 120 min [85]. When larger quantities of ORI-NCs were utilized, such as 34, 84, and 135 g/mL, they significantly reduced cell viability in comparison to the free ORI ($p < 0.05$, $p < 0.01$) [85]. ORI-NCs demonstrated significantly greater endocytosis than free ORI in MDCK cells ($p < 0.01$) [85]. During the transport phase, ORI-NC was taken up by cells in its intact form and expelled from the basolateral membrane of polarized epithelial cells [85].

Sorafenib (SOR) is classified as an oral multi-kinase inhibitor that is utilized to treat advanced HCC [86]. It is responsible for reducing tumor angiogenesis and induces tumor cell apoptosis by suppressing vascular endothelial growth factor (VEGF) [86]. In a study, SOR and parthenolide (PTL) were formulated in combination in the form of a nanocrystal [86]. The purpose of this combination was not only to enhance the poor aqueous solubility of PTL but also to improve the synergistic therapeutic effects with SOR [86]. PTL is a natural sesquiterpene lactone that is derived from feverfew (*Tanacetum parthenium*), which demonstrates notable anti-cancer and anti-inflammatory properties [86]. The SOR and PTL combined nanocrystals (SOR/PTL-NCs) were manufactured by precipitation and high-pressure homogenization method (PHPH) with poloxamer 188 and lecithin as stabilizers [86].

Furthermore, a methylimidazole tetrazolium (MTT) assay was used to investigate the combination therapeutic effects of SOR and PTL-NCs [86]. After 24 h of incubation, the results demonstrated combining SOR and PTL-NCs had greater inhibitory effects [86]. In vitro SOR/PTL-NCs demonstrated a considerably greater inhibitory effect on HepG2 cells in comparison to SOR/PTL, showing that PTL-NCs had enhanced combination therapeutic effects with SOR [86]. The SOR/PTL-NCs displayed excellent synergistic therapeutic effects in comparison to solitary SOR and PTL, with a tumor inhibition rate of 81.86% [86]. A summary of the nanocrystals and NCC used in the treatment of antineoplastic diseases as shown in Table 3.

**Table 3.** Summary of the nanocrystals and NCC in the treatment of antineoplastic diseases.

| Anti-Cancer Agent | ROA | MoM | Stabilizers | Animal Model/Cell Model | Observations | Ref. |
|---|---|---|---|---|---|---|
| Paclitaxel Nanocrystals | Oral Route | HPH | | Male Wistar rats weighing $250 \pm 20$ g | • A higher AUC of 10 folds than the traditional paclitaxel solution. | [87,88] |
| Paclitaxel–disulfiram NCC | Parenteral route | AP | Polyvinyl pyrrolidone (PVP) | Human lung adenocarcinoma A549 cells and Taxol resistant Taxol cells were used in vitro. | • Paclitaxel disulfiram NCC had a drug loading capacity of approximately 43% compared to 36% in paclitaxel nanocrystals.<br>• Paclitaxel–disulfiram NCC in A549/Taxol cells had an uptake of 14-fold higher than that of paclitaxel nanocrystals.<br>• Improved tumor inhibition for breast cancer was demonstrated with paclitaxel NCC. | [83] |
| Paclitaxel folate nanocrystals (PTX-folate) | Oral route | HPH | Pluronic® F-127 | Human carcinoma cell line | • Cytotoxicity was reduced from 10% to 5% in targeted PTX (PTX-folate nanocrystals) compared to non-targeted PTX nano crystals. | [79] |
| Bexarotene nanocrystals | Oral Route | MF&AP | Lecithin and Pluronic® F-68 | Wistar rats of body weighing $250 \pm 20$ g | • The $C_{max}$ of Bexarotene nanocrystals ($2.50 \pm 0.35$ µg/mL) is lower than that of Bexarotene solution ($5.29 \pm 0.97$ µg/mL).<br>• Lower $C_{max}$ and higher AUC indicate a decrease in side effects and the high indicated an increase in bioavailability due to an increase in bioavailability.<br>• Enhancement in vivo and in vitro antitumor activity in A549 bearing mice. | [79] |
| Baicalein–nicotinamide NCC | Oral route | HPH | Poloxamer 188 | Sprague–Dawley rats weighing $250 \pm 20$ g | • BE NCT NCC, BE-NCT co-crystals, and BE nanocrystals showed a 2.17-fold, 2.01-fold, and 1.74-fold increase in 360 min dissolution rates in comparison to BE coarse powder. | [68] |
| Etoposide Nanocrystal suspension | Parenteral route | AP | Pluronic® F-127 | Mice model | • ETO nanocrystal suspension demonstrated a sustained release drug profile kinetics.<br>• Greater anticancer efficacy in vivo compared to the marketed form.<br>• Prolonged stability for months in its crystalline form. ETO nanocrystals proved to be more effective and safer than the marketed form. | [81] |

**Table 3.** *Cont.*

| Anti-Cancer Agent | ROA | MoM | Stabilizers | Animal Model/Cell Model | Observations | Ref. |
|---|---|---|---|---|---|---|
| Curcumin nanocrystals attenuate cyclophosphamide | Parenteral route | AP | | Swiss albino mice induced testicular toxicity | • Administration of curcumin nanocrystals successfully alleviated the cyclophosphamide-induced testicular toxicity and enhanced sperm functional competence.<br>• The alleviating effect of nano crystals was measured in testicular tissue by inhibiting cyclophosphamide-induced DNA damage and oxidative stress. | [89] |
| Chondroitin sulphate modified doxorubicin nanocrystals | Parenteral route | SE | | Cancer cells were used in vitro | • A high drug loading content of approximately to 70% was observed with doxorubicin nanocrystals compared to 18% in doxorubicin micelles. | [90] |
| | | AP | | | | [91] |
| Oridonin (iv) nanocrystals (ORI-NCs) | Parenteral route | AP | Polyvinyl pyrrolidone | There was no animal model instead MDCK cells were used in vitro | • ORI-NCs had a higher dissolution rate than pure ORI in 120 min. ORI-NCs substantially reduced cell viability when compared to free ORI at elevated concentrations (34, 84, and 135 g/mL). In MDCK cells, ORI-NCs showed significantly greater endocytosis than free ORI ($p < 0.01$). | [85] |
| Sorafenib parthenolide nanocrystals (Sora/PTL-NCs) | Parenteral route | HPH | Poloxamer 188 | Female nude mice model | • In vitro, the combined therapy of Sora and PTL-NCs (Sora/PTL-NCs) had superior therapeutic efficacy on intracellular uptake, cell proliferation inhibition, and migration inhibition than either PTL or Sora individually.<br>• An antitumor effect of 81.86% was obtained with Sora/PTL-NCs compared with Sorafenib (58,8%) and Parthenolide (48.84%) | [86] |
| Rapamycin nanocrystals (Rapumune®) | Oral route | WMM | Poloxamer 188 & Povidone | Rapamycin mouse model | • A 21% higher bioavailability was observed in Rapumune® in comparison to Sirolimus® | [92,93] |

**Table 3.** *Cont.*

| Anti-Cancer Agent | ROA | MoM | Stabilizers | Animal Model/Cell Model | Observations | Ref. |
|---|---|---|---|---|---|---|
| Docetaxel Nanocrystals surface-modified with Herceptin® (HCT-DTX-NCs) | Parenteral route | SE | Tween® 80 | Human lung cancer cell line, MCF cells | • In PBS (pH 7.4), HCT-DTX-NCs enhanced drug release compared with DTX-NCs and DTX (pure) containing Tween 80 (0.5% w/v). HCT-DTX-NCs also increased cellular uptake and cytotoxicity in comparison to DTX-NCs and DTX (pure) in MCF-7 cells. | [94] |
| Docetaxel nanocrystals modified with apo-Transferrin human (Tf) (Tf-DTX-NCs) | Parenteral route | AP | Transferrin | A549 cells | • Tf-DTX-NCs substantially enhanced the cytotoxicity and cellular uptake of DTX in the A549 cell line.<br>• At docetaxel concentration of 100 µg/mL, Tf-DTX-NCs (82.6% ± 0.8%) demonstrated a higher cytotoxicity than DTX-NCs (77.4% ± 4.1%) and DTX (pure; 20.1% ± 4.6%) during a 72 h treatment. | [84] |
| Campothecin nanocrystals | Parenteral route | AP | Hyaluronic acid | CD44 positive cancer cells | • HA-coated CPT nanocrystals had considerably improved anticancer activity in treating CD44 overexpressed cancer cells when compared to crude CPT and CPT nanocrystals, which is due to their targeted delivery and accelerated absorption via CD44-mediated endocytosis. | [95] |
| Campothecin nanocrystals | Parenteral route | AP | Boric acid | Human cervical carcinoma Hela cells and Human carcinoma A549 | • Boric acid coated nanocrystals of camptothecin, exhibited improved cytotoxic activity (IC50 < 5.0 µg/mL) to cancer cells in comparison to synthetic polymer-coated CPT nanocrystals and free CPT. | [96] |

AP—antisolvent precipitation, HPH—high pressure homogenization, MoM—method of manufacture, ROA—route of administration, SE—solvent evaporation, WMM—wet media milling.

## 5. Regulatory Limitations of Crystalline Products

There are co-crystal products that have been marketed since 2009. A summary of available co-crystals, their composition, indication and status as depicted in Table 4. Formulations such as Suglat® (Ipragliflozin L-Proline) (Astellas Pharma and Kotobuki Pharmaceutical, Japan), Entresto® (sacubitril and valsartan) (Novartis, Switzerland), Steglatro® (ertugliflozin) (Merck Sharp and Dohme B.V, Netherlands), and Steglujan® (ertugliflozin and sitagliptin) (Pfizer, USA) were marketed between the years 2014 to 2017 [61,97].

**Table 4.** Summary of co-crystal formulations presently marketed.

| Co-Crystal | Composition | Indication | Status | Ref. |
|---|---|---|---|---|
| Seglentis® | Tramadol–celecoxib (1:1) | Acute postoperative pain | Marketed (2021) | [98] |
| Imbruvica® | Ibrutinib | Chronic lymphocytic leukemia | Marketed (2021) | [99] |
| Steglatro® | Ertugliflozin | Type-2 diabetes mellitus | Marketed (2017) | [56,61] |
| Steglujan® | Ertugliflozin and Sitagliptin | Type-2 diabetes mellitus | Marketed (2017) | [61,97] |
| Beta-chlor® | Chloral hydrate and betaine | Sedation | Marketed (2016) | [56,97] |
| Entresto® | Sacubitril and Valsartan | Used for reducing the risk of heart failure | Marketed (2015) | [56,61] |
| Suglat® | Ipragliflozin L-Proline | Used in the treatment of diabetes mellitus type 2 | Marketed (2014) | [60,65,100] |
| Lexapro® | Escitalopram oxalate | Depression | Marketed (2009) | [56,98] |
| Dramamine® | Diphenhydramine and 8-chlorotheophylline | Prevention of motion sickness (nausea and vomiting) | Marketed (1972) | [25] |
| Depakote® Epilim and Divalproex sodium | Valproic acid exists as an acid form and a sodium salt (sodium valproate) form whereas the co-crystal form contains both valproic acid and sodium valproate | Epilepsy | Marketed (1967) | [26,60,100] |

Some products were identified as co-crystals at a later stage, while already being on the market, such as Lexapro® (Escitalopram oxalate oxalic acid) (Lundbeck, Denmark) and Beta-chlor® (Chloral hydrate and betaine) (Franklin Laboratories, India) [26,60,101]. TAK 020 co-crystal (a tyrosine kinase inhibitor), indicated for the potential treatment of rheumatoid arthritis, is still at its phase I clinical trials [98]. For a drug to be declared co-crystal, one must provide proof that illustrates the presence of an API and co-former in a unit cell and both API and co-former must have functional groups that are non-ionizable.

The regulatory classification of pharmaceutical co-crystals of the United States Food and Drug Administration (USFDA) further outlines that ΔpKa (pKa (conjugate acid of base)—pKa (acid)) should be less than 1, resulting in negligible proton transfer and co-crystals formation, because of the non-ionic interaction. However, if it is determined that the classification of a pharmaceutical solid as a salt or co-crystal is not based on these relative pKa values, the use of spectroscopic instruments and other orthogonal devices can be used to provide proof otherwise [102].

Physicochemical and mechanical properties of a co-crystal formulation should be investigated prior to selecting a suitable co-crystal [100]. Melting point, hygroscopicity, solubility, hardness, plasticity, and elasticity are all examples of physical properties of solid-

state materials [101]. Hygroscopicity of a drug impacts the physicochemical properties such as solubility, dissolution rate, stability, bioavailability, and mechanical properties [102]. Investigation must therefore indicate whether there is an effective method for increasing drug substances' physical properties and ensuring their physical stability. Chemical deterioration of pharmacological components occurs often throughout manufacturing and storage, making it difficult to create compatible pharmaceutical formulations [103]. The co-crystal should overcome API chemical instability in the solid form. Elasticity, plasticity, viscoelasticity, and fragmentation mechanisms are among the mechanical deformation mechanisms for solid materials [101]. Many organic chemicals, on the other hand, have poor mechanical characteristics, making tablet formation difficult [102]. The co-crystal candidate should be able to overcome these hurdles to maintain the quality, safety, or efficacy of the formulation [102]. The safety of the co-former comes into question when selecting a co-crystal candidate; the co-former should be safe and nontoxic in the amount required for administration of therapeutic doses of the drug [100]. One of the hurdles faced in co-crystal drug formulation is the challenge associated with the designing and synthesis process, because there is no guarantee that the synthesized co-crystal is pharmaceutically acceptable to provide potential benefits [25]. Furthermore, the safety of co-formers, unpredictable performance during dissolution and solubility studies, difficulties in establishing in vitro to in vivo correlation (IVIVC), as well as polymorphism act as major stumbling blocks in the development of co-crystals. With polymorphs, their melting points and solubilities differ, affecting the dissolution rate and thereby the bioavailability of the drug in the body [100,104].

The regulatory classification of pharmaceutical co-crystals was first done by the FDA, which determined their development and quality control strategies. The FDA, Centre for Drug Evaluation and Research (CDER) classified co-crystals as 'drug product intermediates' (DPIs) which was undesirable for the industries at the time because co-crystals as DPIs would require different regulatory reporting requirements, unlike polymorphs or salts [105]. The regulations imposed on pharmaceutical co-crystals are like those of polymorphs of an API because the solvates are of the initial drug substance, hence they are not regarded as new API. Solvates are multicomponent crystalline solid molecules made up of an API, excipient, or solvent and a substance formed from that solvent. Like hydrates, solvates are subclasses of co-crystals and are commonly referred to as pseudopolymorphs [106,107]. They can be used to expand a co-crystal's number of related solid forms. Solvated crystals are said to have improved solubility, bioavailability, and dissolution rate; however, they are less stable and may dissolve during storage [43,108]. Pharmaceutical industries benefit from producing co-crystals at existing formulation facilities using APIs and co-formers without extra requirements of current good manufacturing practice (cGMPs) [105].

Drug manufacturers are still required to submit the relevant data for new drug applications (NDAs) and abbreviated NDAs (ANDA) containing a co-crystalline form supporting the structure of the co-crystals [102]. The European Medicines Agency (EMA) released a report in 2015 that classified co-crystals in a similar way to the salts of APIs [25]. The regulations also classify those co-crystals as eligible for generic application like salts, and for a co-crystal to achieve the status of new active substance (NAS), they should demonstrate the difference in efficacy and/or safety with respect to that of API [109].

Before clinical trials on humans for a co-crystal API are performed, preauthorization must first be obtained from the Food and Drug administration (FDA). An application of an investigational new drug (IND) must be made with the FDA first. This application should indicate the safety profile as well as manufacturing information of the product and a detailed procedure for conducting the clinical trial. The application will then be reviewed by the FDA to ensure that the safety and effectiveness of the drug comply with the method of manufacturing. Once evidence on dissociation of drug from co-crystal before reaching the targeted site of action has been provided, the USFDA will accept the new drug applications [101].

Since co-crystals, hydrates, and solvates are held together by weak interactions that are in most cases broken upon dissolution, when such a form, already authorized as a medicinal product in the EU, is administered orally, it will expose a patient to the same therapeutic moiety. Just as for salts, they will, therefore, not be considered as NASs in themselves unless they are demonstrated to be different with respect to efficacy and/or safety [110].

Under the condition that any difference in, e.g., solubility, lacks any clinical significance, it is possible to include forms with different degrees of hydration (hydrates, including anhydrous forms) as alternatives in the marketing authorization for a single medicinal product. Any such proposal must be justified, and the lack of clinical significance demonstrated, e.g., by comparison of the intrinsic solubility, etc. The relevant sections of the dossier such as manufacturing description and formula, specifications, etc., must consider the actual forms used. The Summary of Product Characteristics (SmPC) may use wording under Section 2 that expresses the content without defining the hydrated state. Different crystal forms of the same composition (polymorphic forms; see Figure 1) may be accepted as alternatives in the marketing authorization for a single medicinal product provided that any chemical or pharmaceutical difference in properties have no clinical significance [110].

The formation of co-crystals, just like salts, is normally subject to compliance with part II of the European Union Good Manufacturing Practice Guide for active substances and ICH Q7. If, however, in more rare cases where a co-crystal is formed in a step during the drug product manufacturing process, such as a wet granulation or hot melt extrusion, the formation falls under part I of the EU GMP Guide (finished product), while part II applies to active component(s) forming the co-crystal [111]. A summary of classifications between USFDA and EMA guidelines as depicted in Table 5.

There are also specific preclinical data that should be generated for products prior to the application to the relevant regulatory authorities that act as benchmarks for the determination of whether a co-crystal has been formulated. New regulatory guidelines from the FDA and EMA aimed at pharmaceutical co-crystal development, which should be facilitated with the requirements and approval procedures [112]. These guidelines indicate the importance of characterization and quality control of co-crystals [20]. During screening, grinding is referred to as the most preferred synthesis method for co-crystals [113]; this is because it has a brief processing time, uses small sample size, and different compounds can be used in the process [20]. Powder X-ray diffraction (pXRD) and thermal analysis have also been used in screening for purposes of characterization [114]. Structural details and the selection of a proper co-former can be predicted by the in situ monitoring tool [115], whereas Raman spectroscopy can be used for quick detection of crystal formation [116] and hot stage thermal microscopy can be used to indicate synthesis of new formations at material interfaces [117]. For structural characterization, the determination of structural and physical properties is a critical step in understanding the quality of co-crystals developed. Many analytical techniques such as Raman spectroscopy (RS), thermal analysis such as differential scanning calorimetry (DSC), hot stage microscopy (HSM), and thermogravimetric analysis (TGA), solid-state nuclear magnetic resonance spectroscopy (NMR), XRD and many more, are used as estimation tools [20]. Quantum chemistry calculations are also used as an approach in estimating interaction energies between molecules, thereby generating more information regarding their supramolecular architecture [118]. Quantum chemistry-based investigations aid in revealing important information about the nature and strength of non-covalent bonds involved in co-crystals [119].

**Table 5.** Summary of classifications between USFDA and EMA guidelines.

| FDA | EMA | Ref. |
|---|---|---|
| Regulatorily categorized as polymorph of the API | Regulatorily categorized as API | [109] |
| Composed of API and another molecule (food or drug co-former) | Composed of an API and a co-former in fixed stoichiometric ratio | [22,106,113] |
| Co-former regarded as an excipient | Co-former regarded as a reagent | [20,109] |
| New chemical entity or new active substance registration is not possible | New chemical entity or new active substance registration possible only if difference in efficacy or safety is proved | [22,106,113] |
| Co-crystal is classified as a polymorph of the API | Co-crystal is classified as similar to the salt of the same active pharmaceutical ingredient | [20,109] |
| US—Drug master Files (DMF)/EMA—Active substance master file (ASMF) registration can be possible but not required | US—Drug master files (DMF)/EMA—Active substance master file (ASMF) registration must be filed | [101,109] |

The single-crystal XRD (scXRD) characterization technique gives a detailed structure of the crystal. This is used for theoretical purposes and for the quality control of co-crystals, whereas the pXRD gives information on the structural aspect of the co-crystals [120], with Fourier transform infrared spectroscopy (FTIR) being used for identifying organic, inorganic, and polymeric materials using infrared light to scan the samples [121]. For characterization of physicochemical properties, TGA and DSC give information on melting temperature, thermal transition crystallinity, as well as hydrate or solvate formation, whereas pXRD can be used to identify physical changes that might have occurred during measurements, whilst chemical analysis of compounds that are altered thermally can be done using the FTIR and Raman spectroscopy. The pXRD can also be used to monitor co-crystal stability. The shake-flask method is used for determining the solubility of API at a certain temperature within a certain media [122]. Animal studies are used to provide information on how the formulation performs. However, the animal study should be relevant to humans [123]. Overall, vibrational spectroscopic techniques are the most advantageous for purposes of interpretation and characterization of co-crystals [20].

Quantum chemistry investigations use computational approaches to provide an in-depth understanding of co-crystal structures, molecular packing motifs, and corresponding intermolecular interactions [119]. Computational work is also used to determine the optimized geometry of the molecules using different methods [124]. Density functional theory (DFT) and coupled cluster (CC) or quantum Monte Carlo methods are used as standard methods for the quantitative study of large weakly correlated systems [125]. DFT computations are essential in the identification and interpretation of the relationship between the structure and stability or other physicochemical properties of a co-crystal [119]. The Hartree–Fock (HF) method uses spatial coordinate positions of crystals from X-ray structural analysis as its initial coordinates to optimize the geometry of the molecules [124]. CLP-PIXEL method is used to estimate the energy of separate intermolecular interaction between two molecules as well as their lattice energy [125]. Quantum chemistry algorithm based on tree tensor network states (QC-TTNS) further assist in solving problems in quantum chemistry that are intractable by standard techniques such as DFT or CC [125]. Furthermore, the stable relationship between two co-crystals and the weak interactions between APIs and CCFs can be analyzed and compared using the quantum chemistry theory, including, among others, the Hirshfeld surface, the molecular electrostatic potential (MEP) surface, the reduced density gradient (RDG), the quantum theory of atoms in molecules, the frontline molecular orbital, and lattice energy [126].

## 6. Conclusions and Future Perspectives

With cancer deaths continuing to rapidly increase, it is imperative that better treatment regimens are created to find a lasting solution for this deadly disease. Strategies that will improve API pharmacological action are of paramount importance in cancer treatment. The emergence of crystal engineering is a promising method in addressing physicochemical challenges associated with drug delivery systems, through the manufacturing of co-crystals and nanosized crystalline material. These crystalline materials improve API solubility, stability and, in some cases, bioavailability. This in turn improves the pharmacological activity of the API which is critical in disease treatment.

The selection process of an appropriate CCF is imperative in all instances, such as drug–drug co-crystallization, as it results in the success of the crystal engineering technology. Therefore, selecting a CCF that can have a synergistic effect with the API is highly recommended. The selection of appropriate stabilizers can also influence the performance of the technology. However, not all drug–drug co-crystals result in increased physicochemical properties: some result in antagonistic effects. Factors such as crystal size and the crystallization method used must be considered when formulating a crystalline material. The proper characterization techniques should also apply, to understand the nature of the crystalline material.

Although crystalline materials have shown great promise in the treatment of cancer both in vivo and in vitro, most of these materials have not been authorized to be used in clinical trials or the subsequent roll out to market authorized medicines. This is due to the strict regulations that are imposed on drug manufacturers with regards to NDAs and ANDA. This challenge can be overcome if regulations are eased to motivate drug manufacturers. The USFDA and EMA have put in place different regulatory considerations regarding co-crystals; the USFDA classifies them as polymorphs of an API whereas the EMA classifies them as salts of an API. Manufacturers are still required to submit the relevant data for new drug applications (NDAs) and abbreviated NDAs (ANDA) containing a co-crystalline form supporting the structure of the co-crystals.

There are some challenges faced in co-crystal drug formulation, for example, there is no guarantee that the synthesized co-crystal is pharmaceutically acceptable to provide potential benefits, the safety of their co-formers has unpredictable performance during dissolution and solubility studies, difficulties are encountered in establishing IVIVC, and polymorphism can act as a major stumbling block in the development of co-crystals. There is a need for revisitation of regulation to motivate more research to be conducted around crystalline materials and more formulations to be released for market use.

There is also a need to develop other co-crystal material to be used in the diagnosis of cancer or for theranostic purposes. Furthermore, there is a dearth of knowledge and indeed potential of nano(co-)crystalizing radiopharmaceuticals that are essential for theranostic purposes in neoplastic diseases. There remains the potential for further modification in delivery methods for these crystalline materials by use of stimuli responsive carriers such as hydrogels. Overall, crystal and/or nano engineering remain hugely underutilized despite their great potential in the treatment of cancer and remain a promising approach in the treatment of neoplastic disease that needs to be translated into authorized medicines for use in clinical practice.

**Author Contributions:** Conceptualization, B.A.W. and M.S.P.; writing—original draft preparation, E.M.K., L.N.K., T.V.C., S.M., E.B., M.M., K.D.K., M.S.P., P.H.D. and B.A.W.; writing—review and editing, E.M.K., L.N.K., T.V.C., S.M., E.B., M.M., K.D.K., M.S.P., P.H.D. and B.A.W.; visualization, E.M.K. and B.A.W. supervision, B.A.W., P.H.D., M.S.P., E.B., M.M. and S.M. All authors have read and agreed to the published version of the manuscript.

**Funding:** This research received no external funding.

**Institutional Review Board Statement:** Not applicable.

**Informed Consent Statement:** Not applicable.

**Data Availability Statement:** Not applicable.

**Acknowledgments:** The authors would like to acknowledge Sefako Makgatho Health Sciences University for the article processing charges.

**Conflicts of Interest:** The authors declare no conflict of interest.

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
