# Peer review of "Nano- and Crystal Engineering Approaches in the Development of Therapeutic Agents for Neoplastic Diseases"

_crystals, doi:10.3390/cryst12070926_

Round 1

Reviewer 1 Report

The Authors should really improve the presentation style. This applies especially to the figures that are not very informative and too simple.

Line 15 “it can lead to increased doses, leading to the possibility of increased”-this doesn't sound good

Lines 92-113, this part is really not needed. This review focuses on the crystals and their role in the therapy, not on the physiology and pathophysiology of tumor.

Lines 143-145, I don’t agree with this statement. You can achieve the similar results by looking for another polymorphic form of a single component crystal as well.

Lines 149-152, but, on the other hand, the amorphous forms often possess higher solubility and better dissolution kinetics

Lines 158-161, this sentence is grammatically incorrect

Line 169, this should be moved next to the Figure 2.

Table 3, the number of examples (only 7) is really small. Haven’t you find and more in the literature?

Lines 375-380, what were the compositions of those formulations? Were they therapeutic agents for neoplastic diseases? If not, why they are mentioned in this review?

The authors should highlight the differences between the co-crystals and solvates. For example, the ethanolic solvate of an API can be also called a co-crystal, according to the Author’s definition.

Lines 478-483, the Authors do not mention important methods of studying co-crystals that are based on quantum chemical calculations. Such molecular modelling studies can be very beneficial.

At the end, the part describing each author’s individual contribution is missing. This is mandatory in Crystals.

Author Response

Dear Reviewers,

Thank you for the comments concerning our manuscript titled

Nano- and Crystal Engineering Approaches in the Development of Therapeutic Agents for Neoplastic Diseases”. All your comments were valuable and very insightful for revising and improving our paper to the current version, as well as the important guiding significance to our research. We have studied the comments carefully and have made corrections and hope that our updated manuscript will gain your approval. The revisions made are all tracked and marked in the paper.

Thanking you in advance for the consideration that you have given to the manuscript.

Sincerely yours,

Emmanuel Kiyonga

REVIEWER 1

  1. The Authors should really improve the presentation style. This applies especially to the figures that are not very informative and too simple.

Thank you for the comment. We have made changes to this section by adding alternative figures that have more information.

  1. Line 15 “it can lead to increased doses, leading to the possibility of increased”-this doesn't sound good

Thank you for picking this up. This sentence has been reconstructed and rephrased. to make it read as “which can lead to low drug absorption, increased doses and subsequently poor bioavailability and the occurrence of more adverse events”

 Lines 92-113, this part is really not needed. This review focuses on the crystals and their role in the therapy, not on the physiology and pathophysiology of tumor.

We appreciate your view on this part. We have carefully considered your suggestion and we thought that it would be advantageous to the reader if we left this section included because it introduces the reader to the nature of neoplastic disease we are speaking of before we described the treatment options. Furthermore, noting the comment of reviewer 2 who suggested that we include a figure regarding the predominance of cancer types to improve our manuscript. This we have done as per the recommendation of reviewer 2

  1. Lines 143-145, I don’t agree with this statement. You can achieve the similar results by looking for another polymorphic form of a single component crystal as well.

Thank you for your comment. We have made deletions on this sentence. We have removed the statement: “Unlike single component crystals”.

  1. Lines 149-152, but, on the other hand, the amorphous forms often possess higher solubility and better dissolution kinetics

Thanks for this comment. We have made additions to this sentence to make it read as “The amorphous solid-state form has higher Gibbs free energy, internal energy, specific volume, solubility, and thus dissolution rate compared to co-crystals.”

  1. Lines 158-161, this sentence is grammatically incorrect

Thanks for identifying this error, we have fixed this sentence by deleting all of it and replacing it with a more grammatically sound one which reads as “Molecules that are already polymorphic and molecules that can adopt other packing patterns while still fulfilling the needs of the hydrogen-bond acceptor or donor existing on the two components, should be employed during co-crystallization.” 

  1. Line 169, this should be moved next to the Figure 2.

Thank you for identifying this. It has been noted and fixed accordingly.

  1. Table 3, the number of examples (only 7) is really small. Haven’t you find and more in the literature?

Thank you for your gracious suggestion. We have added more examples from literature to the table as suggested.

  1. Lines 375-380, what were the compositions of those formulations? Were they therapeutic agents for neoplastic diseases? If not, why they are mentioned in this review?

Thank you for this comment. Below are the formulations with their compositions as added to the paper.

Suglat (Ipragliflozin L-Proline)

Entresto (sacubitril and valsartan)

Steglatro (ertugliflozin)

Steglujan (ertugliflozin and sitagliptin)

Thank you for pointing this out. Indeed, these formulations are not therapeutic agents for neoplastic disease. They are instead indicated for other conditions such as type 2 diabetes mellitus and heart failure in adults. The reason why they are mentioned is because they are all co-crystal products, and since the section within which they are mentioned is speaking of the regulatory limitations of co-crystalline materials, there was a need to mention some formulations currently on the market that are co-crystalline materials. On this section, focus was not put only on co-crystals used in neoplastic diseases but on available co-crystals irrespective of the indication. The same regulations would apply to any co-crystal formulation on the market irrespective of their indication. This is the basis on which these drugs were included in this review. We have also included more examples on this section from literature to substantiate the point further.

  1. The authors should highlight the differences between the co-crystals and solvates. For example, the ethanolic solvate of an API can be also called a co-crystal, according to the Author’s definition.

Thank you for the comment. The differences have been highlighted as suggested.

  1. Lines 478-483, the Authors do not mention important methods of studying co-crystals that are based on quantum chemical calculations. Such molecular modelling studies can be very beneficial.

Thank you very much for this critical point. We have included this section as suggested and indeed, including it was very critical for improving our manuscript. Methods for studying co-crystals based on quantum chemical calculations have been added.

  1. At the end, the part describing each author’s individual contribution is missing. This is mandatory in Crystals.

Thank you for this gracious reminder. We have indeed added this mandatory section as recommended.

Reviewer 2 Report

In the work “Crystal and Nanoengineering Approaches in the Development of Therapeutic Agents for Neoplastic Diseases” the authors reviewed the development and use of crystalline formulations for the treatment of various neoplastic diseases and how they can be effectively applied in clinical practice.

The manuscript is well written and easy to follow. However, there is space for improvement:

1 – Please improve the abstract section.

2 – Lines 76-78 – please revise sentence

3- line 92 – authors could include a Figure regarding the predominance of cancer types. This will improve the manuscript.

4  - line 169 – Figure 2 caption should be with Figure 2. Please revise

5 – Improve the quality of all figures to increase the overall quality of the manuscript

Author Response

Dear Reviewers,

Thank you for the comments concerning our manuscript titled “Crystal and Nanoengineering Approaches in the Development of Therapeutic Agents for Neoplastic Diseases”. All your comments were valuable and very insightful for revising and improving our paper to the current version, as well as the important guiding significance to our research. We have studied the comments carefully and have made corrections and hope that our updated manuscript will gain your approval. The revisions made are all tracked and marked in the paper.

Thanking you in advance for the consideration that you have given to the manuscript.

Sincerely yours,

Emmanuel Kiyonga

Below are the corrections made in the paper and the responses to the reviewers’ comments.

REVIEWER 2

  1. Please improve the abstract section.

Thank you very much for this comment. We have indeed revised the abstract and changes have been made as suggested.

  1. Lines 76-78 – please revise sentence

Thank you for pointing this out. The sentence has been revised and additions have been made. The sentence now reads as “Nanosystems increase the concentration of the drug selectively to the target site, while reducing side effects associated with wide drug distribution. Therefore, nanosystems were found to reduce toxic effects brought by antineoplastic agents”

  1. line 92 – authors could include a Figure regarding the predominance of cancer types. This will improve the manuscript.

Thank you for this recommendation. We have included a figure regarding the predominance of cancer types as suggested.

  1. line 169 – Figure 2 caption should be with Figure 2. Please revise

Thank you for pointing this out. We have revised the section as recommended.

  1. Improve the quality of all figures to increase the overall quality of the manuscript

Thank you for the comment. We have made changes to this section by adding alternative figures that have more information and removing figures that compromised the quality of our manuscript.

Round 2

Reviewer 1 Report

The Authors have corrected their manuscript. This version can be accepted.